# Dietary Sodium Consumption and 3-Year Progression of Subclinical Arterial Damage in Adults with Cardiovascular Risk Factors

**DOI:** 10.3390/nu17050808

**Published:** 2025-02-26

**Authors:** Eirini D. Basdeki, Kalliopi Karatzi, Yannis Manios, Petros P. Sfikakis, Antonios A. Argyris, Athanase D. Protogerou

**Affiliations:** 1Cardiovascular Prevention & Research Unit, Clinic & Laboratory of Pathophysiology, Department of Medicine, National and Kapodistrian University of Athens, 11527 Athens, Greece; eirinibasdeki@gmail.com (E.D.B.); and1dr@gmail.com (A.A.A.); 2Laboratory of Dietetics and Quality of Life, Department of Food Science & Human Nutrition, Agricultural University of Athens, 11855 Athens, Greece; pkaratzi@aua.gr; 3Department of Nutrition and Dietetics, School of Health Science and Education, Harokopio University, 17671 Athens, Greece; manios@hua.gr; 4Institute of Agri-food and Life Sciences, University Research & Innovation Center, H.M.U.R.I.C., Hellenic Mediterranean University, 71003 Crete, Greece; 5European Centre for Obesity, Harokopio University, 17671 Athens, Greece; 61st Department of Propaedeutic Internal Medicine, Medical School, National and Kapodistrian University of Athens, 11527 Athens, Greece; psfikakis@med.uoa.gr

**Keywords:** dietary sodium intake, cardiovascular disease, subclinical vascular damage, progression, arterial stiffness, PWV, atheromatosis, carotid plaques, femoral plaques

## Abstract

**Background/Objectives**: Available data regarding associations between sodium (Na) intake and biomarkers of subclinical arterial damage (SAD) are scarce. This study aimed to investigate the possible associations between Na intake and the 3-year progression of SAD in subjects with cardiovascular disease (CVD) risk factors. **Methods**: Participants underwent CVD risk assessment, vascular assessment [arterial stiffness by pulse wave velocity (PWV), and atheromatosis, as the existence of carotid and/or femoral plaques], anthropometric measurements [at baseline and 3-year follow-up (FU)], and dietary assessment at FU. **Results**: A total of 675 adults (47.9% males, 55.02 ± 13.79 years) were included. Na daily consumption quartiles (Qs) ranged from very low consumption in Q1 (811.72 ± 241.81 mg) up to twice the recommendations in Q4 (3487.92 ± 1025.92 mg). No statistically significant associations were observed between Na intake and changes in SAD biomarkers, after adjustment for age, sex, presence of hypertension, presence of dyslipidemia, smoking, mean arterial pressure, BMI, chronic inflammatory diseases, and energy intake. The results remained the same, even after the assessment of misreporting and the correction of Na intake. **Conclusions**: Dietary Na intake was not significantly associated with changes in PWV and carotid or femoral plaques, even in the high Q that was twice as high as the recommended intake. Research in different additional adult cohorts is needed to further investigate whether Na consumption independently affects vascular health.

## 1. Introduction

Globally, the sodium (Na) intake in adults exceeds the recommended by the World Health Organization (WHO) intake of 2000 mg/day, and is estimated to be close to 4300 mg daily [1]. Higher intakes than the recommended dietary Na intake are strongly associated with the risk of a variety of non-communicable diseases, including high blood pressure (BP) and cardiovascular disease (CVD) [2].

The onset of subclinical arterial damage (SAD) begins decades before the development of established CVD [3]. Early detection of either functional or structural SAD can lead to timely diagnosis and the prevention of CVD, and can be assessed by a wide variety of non-invasive vascular biomarkers [3,4]. Pulse wave velocity (PWV) measured by applanation tonometry is the most widely accepted index of arterial stiffness, whereas, atheromatosis is commonly assessed using ultrasonography for the detection of carotid and/or femoral plaques [3].

The association of Na intake with SAD has been investigated in a few studies with the purpose of identifying potential pathways that might explain the increased incidence (or prevalence) of CVD in populations with higher than recommended Na intake. We recently conducted a systematic review of all available studies (observational and interventional) that investigated the associations between Na intake and SAD [5]. We concluded that the available evidence was not enough to support a direct and causal association between Na and SAD, mainly owing to the fact that the available data derive mostly from small, heterogeneous, and not well-designed studies [5]. To our knowledge, there are so far only two observational studies that have examined the associations between dietary Na intake and the progression of SAD. Both available studies examined the change (Δ) in arterial stiffness (cfPWV and baPWV), in either healthy adults or patients with chronic kidney disease (CKD), providing conflicting conclusions, as both positive and inverse associations between Na intake and ΔPWV were found [6,7]. However, these studies had several limitations, including small sample size, short period of follow-up, and different types of examined vascular biomarkers [6,7].

In the present study, we aimed to examine the potential association between dietary Na intake and the 3-year progression in aortic stiffness (carotid to femoral, cfPWV) and subclinical atheromatosis (carotid and femoral) in adults free of established CVD but with CVD risk factors.

## 2. Materials and Methods

### 2.1. Study Design & Population

The present study was conducted at the Cardiovascular Research Laboratory of the First Department of Propaedeutic and Internal Medicine, Laiko General Hospital, Athens, Greece. Participants consisted of adults free of established CVD (defined as pre-existing coronary artery disease, stroke, or peripheral arterial disease) but with CVD risk factors, who consecutively visited the laboratory for SAD assessment and CVD reclassification. All participants were examined at two time points: at baseline and at 3 years follow-up (FU) visit. In both visits, participants underwent the same exams (CVD risk assessment, vascular assessment, and anthropometric measurements) in addition to dietary assessment at FU. The present protocol was conducted according to the World Medical Association Declaration of Helsinki and was approved by the Laiko’s Hospital ethics committee [8]. All participants provided written informed consent.

### 2.2. Definition of CVD Risk Factors and Chronic Inflammatory Diseases

Hypertension was defined by the use of antihypertensive drugs and/or office BP measurement higher than 139/89 [average of 3 sequential readings with 1 min interval in the supine position after at least 10 min of rest (MicrolifeWatchBP Office, Microlife AG, Widnau, Switzerland)] [9]. The definition of dyslipidemia used was lipid-modifying drugs and/or low-density lipoprotein cholesterol levels >160 mg/dL [10]. Current smoking was considered as the daily use of at least 1 cigarette, every day of the week, and ex-smoking was considered as discontinuation for at least 6 consecutive months or more. Chronic inflammatory diseases were defined as the presence of any of the following: rheumatoid arthritis, ankylosing spondylitis, psoriatic arthritis, and HIV.

### 2.3. Vascular Assessment

Vascular assessment was performed by the same examiner. BP was assessed as described above. PWV was assessed by a common tonometric method and the use of a generalized transfer function (Sphygmocor; Atcor Medical, Sydney, Australia) [11]. IMT and the presence of plaques (in the carotid and femoral arteries) were assessed by high-resolution ultrasonography (Vivid 7 Pro, General Electric Healthcare, Little Chalfont, UK). IMT was calculated adjacent to any plaque (if present) at the far wall of the left and right common carotid artery, and the average measurements of two sequential images were used. Atheromatic plaque was defined as a local increase in the IMT > 50%, compared to the adjacent vessel wall or a bulging to the lumen IMT > 1.5 mm. The presence of any plaque at the (left or right) common carotid arteries or carotid bulbs was considered as carotid atheromatosis and the presence of plaque at the (left or right) femoral arteries was considered as femoral atheromatosis [12].

### 2.4. Anthropometric Measurements

Participants underwent body weight and height measurements. Body weight was measured by using the weight scale Tanita Body Composition Analyzer (BC-418) and height by stadiometer (SECA 213); both were used according to the manufacturer’s instructions. Body mass index (BMI) was calculated as weight (Kg)/height (m^2^).

### 2.5. Assessment of Dietary Na Intake

Two 24 h dietary recalls (one weekday and one weekend day) were conducted on all the study’s participants by trained dieticians through telephone interviews. The recalls were conducted with at least one-week intervals. Collected data were analyzed in macro- and micro-nutrients, as well as in energy intake, by using the Nutritionist Pro Software (Axxya Systems Nutritionist Pro TM 2011). The Nutritionist Pro food database was expanded by adding analyses of traditional Greek foods and recipes. The calculation of total daily Na intake was conducted as follows: (i) by estimating the mean amount of Na derived from the dietary recalls using the Nutritionist Pro Software, and (ii) by estimating discretionary Na intake. Discretionary Na is the amount of salt added during cooking or the use of table salt, which cannot be estimated by dietary recalls. The available data suggest that discretionary Na represents around 15% of total Na intake [1,13]. According to these data, the calculation of discretionary Na in this analysis was conducted by assuming that 15% of the total Na intake was derived from cooking and table salt. Finally, total Na intake was calculated as the sum of the mean estimated Na intake from dietary recalls plus discretionary Na.

### 2.6. Assessment of Dietary Misreporting

Assessment of energy intake misreporting was conducted in order to assess Na intake misreporting. Overreporting (OR) or underreporting (UR) of energy were calculated according to Huang et al. [14] approach. This approach is based on the principle that reported energy intake (EI) and predicted total energy expenditure (TEE) are equal, assuming weight stability; statistical comparison of reported EI and predicted TEE is conducted, and cut-off points are set between ±1 SD. Predicted TEE was calculated by using Schofield’s equation, separately for men and women [15]. More specifically, participants were identified as acceptable reporters, UR, or OR of EI according to whether the individual’s ratio %EI/TEE was within, below, or above the ±1 SD range, respectively. The ±1 SD cut-off points were calculated by the following equation:$$\pm 1SD=\sqrt{\left(\frac{C{V^{2}}_{rEI}}{d}\right)+C{V^{2}}_{pTEE}+C{V^{2}}_{mTEE}},$$
where *CVrEI* stands for the within-person coefficient of the variation in the EI reported, *d* is the number of dietary assessment days, *CVpTEE* is the error in the predicted TEE equations, and *CVmTEE* is the day-to-day variation in TEE, calculated by using the method of doubly labeled water (DLW) [14]. The number of days was d = 2 for the average of two interviews; *CVrEI* and *CVpTEE* were calculated separately for males and females, and *CVmTEE* was set at 8.2%, as estimated from DLW measurements [16]. In this study, participants were defined as acceptable reporters if %EI/TEE was in the range of 70.9 to 129.1 and 74.4 to 125.6 for males and females, respectively.

Regarding Na misreporting, Murakami et al. found that micronutrient misreporting (including Na) followed EI misreporting [17]. They also found that the concentration of Na per 1000 Kcal of the reported EI did not differ statistically from the concentration of Na per 1000 Kcal of TEE. By using the previous knowledge, we adjusted Na intake for TEE and then repeated the analysis by using the corrected Na.

### 2.7. Statistical Analysis

Statistical analyses were conducted by using SPSS v. 21.0 (SPSS Inc., Chicago, IL, USA). The normality of the distribution of variables was tested using the Kolmogorov—Smirnov test. Categorical variables are presented as relative frequencies (%), and normally distributed continuous variables as a mean ± standard deviation (SD). Since Na intake did not follow normal distributions, quartiles were used to conduct the analyses. Moreover, interactions between Na intake and sex, age, or the existence of inflammation and vascular biomarkers were tested. Multiple linear and logistic regression analyses were conducted to determine the independent associations between Δvascular biomarkers, i.e., i. changes between baseline and follow-up, mean duration 3.2 years (ΔPWV, Δplaques total, Δplaques carotid, and Δplaques femoral) and ii. annual change only for PWV (ΔPWV/Δt)] and quartiles of Na intake. Four models were applied to assess the above associations as follows: model 1: adjusted for age and sex; model 1a: additionally adjusted for the interaction between the presence of inflammatory disease with age; model 2: adjusted for age, sex, existence of hypertension, existence of dyslipidemia, smoking, mean arterial pressure and BMI; model 3: adjusted for age, sex, existence of hypertension, existence of dyslipidemia, smoking, mean arterial pressure, BMI, presence of inflammatory diseases and energy intake.

The results are presented as standardized beta coefficients and 95% confidence intervals (CI) (for ΔPWV and ΔPWV/Δt) or as an odds ratio (OR) expressed as ExpB and 95% CIs (for changes in the existence of total plaques, carotid plaques, and femoral plaques). The level of statistical significance was set at *p* ≤ 0.05 for all analyses, except for the interaction analyses, where it was set at *p* ≤ 0.10.

## 3. Results

The present study included 675 adults (47.9% males, 55.02 ± 13.79 years) with CVD risk factors that were followed up for 3.22 ± 0.25 years. Their average daily Na intake was 1922.40 ± 1134.71 mg, with the consumption in the lowest quartile (Q1) being 811.72 ± 241.81 mg and 3487.92 ± 1025.92 mg in the highest (Q4). The detailed descriptive characteristics of the participants are presented in Table 1 and Table 2.

Table 3 presents the changes (Δ: FU − BSL) in subclinical vascular biomarkers (PWV, total plaques, carotid and femoral plaques). Regarding PWV, the annual change was also calculated and presented in the same table. The second Q, compared to all other Qs, had the highest change in all the vascular biomarkers examined in this study, however all Δ were statistically non-significant between quartiles. The first Q had the lowest progression of SAD, almost minimal, yet the differences with other Qs were non-significant.

In Table 4, associations between changes in subclinical vascular biomarkers and Na intake are presented in multivariate analysis. No statistically significant associations were observed after the adjustment for age, sex, presence of hypertension, presence of dyslipidemia, smoking, mean arterial pressure, BMI, chronic inflammatory diseases, and energy intake.

Interaction analysis between the presence of chronic inflammatory disease and Na regarding their effect on SAD progression showed no interaction. Likewise, interaction analysis between sex and Na regarding their association with SAD progression was also tested, but the results were not statistically significant. Interaction analysis between the presence of chronic inflammation with age regarding their potential association with SAD progression was observed. More specifically, associations between the interaction of inflammation and age with ΔPWV, ΔPWV/Δt, and total plaques were statistically significant (B = 0.313 and *p* = 0.092, B = 0.323 and *p* = 0.083, and B = 0.378 and *p* = 0.043, respectively). After this observation, multivariate analysis between changes in the examined subclinical vascular biomarkers and Na intake adjusted for the interaction of inflammation and age was conducted (model 1a), revealing no statistically significant associations (Table 4).

The misreporting of EI is presented in Table 5. The UR of EI was 49.6%, and the OR was 4.4% in the total sample, respectively. After the assessment of misreporting, corrected Na was calculated, and all analyses were repeated; however, results remained unchanged.

## 4. Discussion

In the present study, we aimed to examine the possible association between dietary Na intake and the 3-year change in biomarkers of SAD in adults with CVD risk factors. Our main findings suggest that dietary Na intake is not significantly associated with changes in PWV and carotid or femoral plaques, even in the highest quartile of Na intake that was twice as high as the international recommendation. Moreover, our findings remained the same, even after calculation and correction for Na misreporting.

There are only two available studies examining the above associations that both examined changes in arterial stiffness (baPWV and cfPWV) [6,7]. In the study of Nerbass et al., that included 1607 adults with stage 3 CKD, participants were split into two groups regarding Na intake (>2300 mg Na/day and <2300 mg Na/day). After 1-year FU, the group with low Na intakes had a significant decrease (Δ) in cfPWV compared to the other group [6]. It should be noted that if the present study had a bigger sample, Q1 might also have shown significant results in ΔPWV vs. Q2, since their regression was marginal (*p* = 0.065), indicating that lower Na intake could be beneficial for changes in PWV. Moreover, the difference between the present findings and those of Nerbass et al. could be attributed to the absence of CKD and the younger age of the participants (55.0 ± 13.8 vs. 72.6 ± 9) in the present study. The other available study was conducted by Jung et al. in 2145 healthy adults (59.9 ± 9.1 years old), where Na intake was calculated with the use of the Food Frequency Questionnaire (FFQ) and a 3-day diet record [7]. Mean Na intake was 2538 ± 1416 mg, and it was positively associated with changes in baPWV in 3-year FU. Despite the fact that Na intake of the aforementioned study is between Q3 and Q4 of the present study, and it would be reasonable to expect similar changes in PWV, in the study of Jung et al., results might also be attributed to normal changes that occur in PWV as age increases, especially over 50 years, and not only in Na intake [18]. Particular emphasis should be given in Q4, where the mean Na intake is higher than Na intake in Jung et al., but the mean age of participants is 10 years younger (Q4: 49.4 ± 13.0 vs. 59.9 ± 9.1 years), which could explain the difference addressed between these two groups. Last but not least, an important point that might be the reason for the differences between this study and the available studies is the overall dietary composition of each study’s participants; however, this information is not available in order to verify this possibility. To sum up, there may be two available studies investigating similar associations in order to compare their results with the current study; however, there is great heterogeneity between all of them, since the current study (i) has the smallest study sample (n = 675 vs. n = 1645 or n = 1607), (ii) different methods of Na assessment were used in each study (24 h recalls vs. spot urine collections or FFQ and diet record), but not the gold standard method (24 h urinary collection), (iii) differences in participants health status (adults with CVD risk factors vs. CKD stage 3 patients or healthy adults), and (iv) the duration of follow-up in one of the studies (3 years vs. 1 year).

It should be noted that the mean Na intake (1922.40 ± 1134.71 mg Na) of the total sample was close to the international recommended intake (2000–2300 mg Na daily), which could be a possible explanation for the absence of associations [1]. However, when total Na intake was divided into quartiles, Q1 had a very low Na intake (811.72 ± 241.81 mg Na), and Q4 had almost double the recommended (3487.92 ± 1025.92 mg Na). Not only the maximal intake (Q4) but also the minimal (Q1) would have been expected to reveal associations between Na intake and SAD, if the growing evidence that supports the existence of a J-shaped association between dietary Na intake and CVD is correct, implying that low Na intake might also increase the risk for CVD [2,19]. The fact that the highest quartile did not show any statistically significant change in vascular biomarkers, could also be attributed to the combination of the following reasons: (i) mean sodium intake in this specific quartile was 3487.92 ± 1025.92 mg Na/day, which is almost twice the recommended intake, but is not an exaggerated amount (mean worldwide intake is about 4300 mg/day), (ii) participants in Q4 had better measurements as far as their lipid levels, medications, and plaques are concerned, due to proper medical and lifestyle changes advice that were given after baseline measurements, (iii) participants in Q4 were the youngest (mean age in follow-up:49.43 ± 13.03 years)of the total sample, (iv) had the lowest mean PWV of the total sample (7.96 ± 2.23 m/sec) and (v) the short duration of the follow-up that might be insufficient for vascular damage to appear. Consequently, the possible reduction in sodium intake that participants might have been advised to follow, their age that was between 45 and 50 years until the follow-up, where the normal increase in PWV is slow (and in plaques is even slower) [20], and the fact that the follow-up was not conducted years later, are a very reasonable explanation of the absence of associations. No significant differences between Qs of Na and biomarkers of SAD were found, and changes in biomarkers of SAD were not different between each Q; however, it was observed that the annual increase in PWV was under 0.04 m/sec in Q1 and Q3, where Na intake was low or almost equal to the recommendation, respectively, while the normal range of PWV progression is 0.44–0.50 m/sec per decade in adults with normal or controlled BP levels [18]. On the other hand, in Q2 and Q4 the increase was higher. In Q4, this increase could be attributed to the high Na intake, and in Q2, where Na intake was low, it could be a result of the fact that this Q includes the participants with the highest age, and an increase in age is associated with an increase in arterial stiffness [18,20].

There are no available data regarding Na intake and its association with changes in plaques. In the systematic review that was recently conducted by our team, data regarding atheromatosis were scarce, with high heterogeneity between the studies that could not lead to any conclusions [5]. After this review, Tsirimiagkou et al., conducted a cross-sectional study in 901 adults with CVD risk factors, to examine the associations of Na intake in SAD, including femoral and carotid plaques, and concluded that very low Na intake (751.0 ± 215.5 mg/day) was associated with presence of femoral plaques only in women [21]. In order to expand the existing knowledge, the present study used data from the same dataset as the study by Tsirimiagkou et al.; however, this study found no significant associations in terms of changes in plaques, possibly due to the smaller sample size, since many participants were lost during the 3-year FU and due to slower progression of atheromatosis, that might need more than 3 years to form significant changes in the arterial wall [22].

This study is one of the few available studies that investigate the existence of associations between Na intake and changes in biomarkers of SAD (PWV and carotid or femoral plaques). However, it also has some limitations; due to its cross-sectional design, no causative relationships can be established, and the reverse causality phenomenon might be the reason for the very well-controlled participants (i.e., lipid profile and medication) in the quartile with the highest Na intake (Q4). Furthermore, the evaluation of Na intake was not conducted with the gold-standard method (24 h urinary collection) but with two 24 h recalls, which are based on participants’ memory and are prone to recall bias. Moreover, the use of 24 h recalls does not allow the assessment of discretionary salt (either in cooking or while eating). In order to minimize any possible decline from the real intake, correction for misreporting has been conducted, and the results remained the same. Moreover, it should be noted that nutritional analysis was conducted by Nutritionist Pro, a program based on US food databases, and there may be differences in the concentration of Na compared with Greek products. However, there is a wide variety of Greek foods registered in this program that were used to minimize declinations. Last but not least, it should be mentioned that the follow-up duration may not be sufficient for the detection of changes in plaques, since more time might be needed for plaque formation to occur.

## 5. Conclusions

To conclude, our findings indicate that Na intake is not associated with changes in SAD biomarkers (PWV and carotid or femoral plaques) over 3-years in adults with CVD risk factors, even in the highest Na intake subpopulation. Since no available data regarding plaques progression existed so far, our findings on PWV are in contrast with previous data, and thus future research is clearly needed. This research should focus on different cohorts of CVD and CVD-free subjects to better examine the role of Na consumption in vascular health.

## Figures and Tables

**Table 1 nutrients-17-00808-t001:** Main characteristics of the study population at follow-up stratified by quartiles (Q) of daily Na intake.

	Total (n = 675)	Q1 (n = 169)	Q2 (n = 168)	Q3 (n = 170)	Q4 (n = 168)
Age (years)	55.02 ± 13.79	57.26 ± 13.38	59.19 ± 12.90	54.20 ± 13.95	49.43 ± 13.03
Sex (%, males)	47.9	30.8	36.9	48.8	75.0
Daily Na intake (mg)	1922.40 ± 1134.71	811.72 ± 241.81	1387.44 ± 155.93	2008.09 ± 225.72	3487.92 ± 1025.92
Energy intake (Kcal)	1680.60 ± 624.65	1254.19 ± 365.32	1446.12 ± 333.59	1733.61 ± 407.90	2290.44 ± 737.92
Time between baseline to follow-up (years)	3.22 ± 0.25	3.22 ± 0.27	3.22 ± 0.24	3.23 ± 0.25	3.23 ± 0.22

**Table 2 nutrients-17-00808-t002:** Cardiovascular disease risk factors and subclinical vascular biomarkers at baseline (bsl) and follow-up (FU) stratified by quartiles (Q) of daily Na intake.

Variable Name (Unit) (Number at bsl/FU)	Baseline					Follow-Up				
	Total (n = 675)	Q1 (n = 169)	Q2 (n = 168)	Q3 (n = 170)	Q4 (n = 168)	Total (n = 675)	Q1 (n = 169)	Q2 (n = 168)	Q3 (n = 170)	Q4 (n = 168)
Smoking (%) (675/675)										
Never smokers	39.4	42.6	45.2	38.2	32.1	39.4	42.0	45.2	38.2	32.1
Former smokers	37.6	34.3	29.2	38.8	47.6	33.9	29.0	26.2	35.9	44.6
Current smokers	23.0	23.1	25.6	22.9	20.2	26.7	29.0	28.6	25.9	23.2
BMI (Kg/m^2^) (675/667)	27.46 ± 4.57	27.69 ± 4.58	27.81 ± 4.52	27.02 ± 4.52	27.33 ± 4.66	27.56 ± 4.68	27.77 ± 4.79	27.69 ± 4.36	27.41 ± 4.87	27.39 ± 4.69
SBP (mmHg) (675/674)	129.70 ± 18.50	129.43 ± 17.76	130.31 ± 19.86	128.89 ± 17.95	130.20 ± 18.50	125.15 ± 16.23	125.13 ± 16.74	126.78 ± 17.58	123.88 ± 15.86	124.83 ± 14.60
DBP (mmHg) (675/674)	77.80 ± 10.60	76.95 ± 10.38	78.29 ± 10.50	77.52 ± 10.08	78.44 ± 11.42	74.96 ± 8.47	74.12 ± 8.80	75.42 ± 8.41	74.74 ± 8.19	75.55 ± 8.47
Blood lipids										
Total cholesterol (mg/dL) (595/580)	199.94 ± 41.01	207.15 ± 44.36	206.81 ± 40.14	196.77 ± 38.49	189.09 ± 38.39	194.24 ± 36.04	197.80 ± 35.65	198.36 ± 36.76	192.65 ± 37.94	188.32 ± 33.17
HDL (mg/dL) (568/574)	55.36 ± 17.48	58.16 ± 19.65	58.54 ± 17.80	55.28 ± 16.07	49.63 ± 14.71	56.10 ± 22.21	60.11 ± 35.24	58.62 ± 15.29	54.74 ± 16.28	51.13 ± 14.57
LDL (mg/dL) (566/574)	123.25 ± 34.77	126.66 ± 40.48	128.47 ± 32.92	120.19 ± 33.18	117.86 ± 31.06	117.04 ± 30.31	118.89 ± 32.33	118.79 ± 32.18	115.25 ± 31.15	115.24 ± 25.34
Triacylglycerols (mg/dL) (580/577)	112.04 ± 56.24	119.82 ± 66.52	111.74 ± 45.95	107.16 ± 54.91	109.47 ± 55.40	109.07 ± 63.06	112.70 ± 55.64	104.17 ± 41.18	109.19 ± 56.65	110.38 ± 88.44
Diabetes mellitus (%) (675/675)										
Type 1 DM	4.7	4.1	2.4	5.9	6.5	5.0	4.1	1.2	7.1	7.7
Type 2 DM	12.9	15.4	11.9	12.4	11.9	13.5	14.8	13.7	11.8	13.7
Hypertension (%) (675/515)										
Yes	49.2	54.4	50.6	48.8	42.9	50.1	59.1	51.2	50.8	41.1
Dyslipidemia (%) (675/675)										
Yes	35.4	40.2	41.1	35.3	25.0	44.3	49.7	49.4	46.5	31.5
HTN drugs (675/660)										
Yes	36.7	43.8	40.5	35.9	26.8	42.0	46.0	44.0	44.8	33.1
Inflammatory diseases (675/674)										
Yes	52.0	53.3	59.5	47.6	47.6	51.2	52.1	58.9	47.3	46.4
Arterial stiffness and mean arterial pressure (MAP)										
MAP (mmHg) (675/674)	99.18 ± 12.80	98.57 ± 12.32	99.72 ± 13.39	98.69 ± 12.34	99.76 ± 13.19	95.64 ± 10.44	95.14 ± 10.83	96.58 ± 10.77	94.98 ± 10.29	95.85 ± 9.87
PWV (m/s) (675/675)	8.38 ± 2.22	8.61 ± 2.26	8.56 ± 2.16	8.38 ± 2.19	7.96 ± 2.23	8.52 ± 2.14	8.61 ± 2.07	8.83 ± 2.22	8.48 ± 2.10	8.17 ± 2.14
Existence of plaques (%) (670/675)										
Yes	50.6	56.8	55.7	45.0	44.8	58.1	62.1	65.5	51.8	53.0
Existence of carotid plaques (%) (667/675)										
Yes	36.7	44.6	40.7	32.1	29.3	44.9	51.5	52.4	38.2	37.5
Existence of femoral plaques (%) (667/675)										
Yes	36.7	40.5	38.3	33.3	34.8	43.7	46.7	46.4	38.2	43.5

BMI: body mass index; SBP: systolic blood pressure; DBP: diastolic blood pressure; PWV: pulse wave velocity; DM: diabetes mellitus; HTN: hypertension.

**Table 3 nutrients-17-00808-t003:** Vascular biomarkers from baseline (BSL) to follow-up (FU) defined as change (Δ: FU − BSL).

	Change from Baseline to Follow-Up (n = 675)			
	Q1 (n = 169)	Q2 (n = 168)	Q3 (n = 170)	Q4 (n = 168)
ΔPWV (m/sec)	0.0001 ± 1.30	0.27 ± 1.47	0.11 ± 1.23	0.21 ± 1.29
ΔPWV/Δt ((m/sec)/year)	−0.002 ± 0.41	0.08 ± 0.45	0.03 ± 0.39	0.06 ± 0.40
Δplaque total (%)				
Decrease	0.0	0.6	0.6	0.6
No change	94.7	88.6	91.7	89.7
Increase	5.3	10.8	7.7	9.7
Δplaque carotid (%)				
Decrease	0.6	0.6	0.0	0.6
No change	92.3	86.8	94.0	89.6
Increase	7.1	12.6	6.0	9.8
Δplaque femoral (%)				
Decrease	0.0	0.0	0.6	0.0
No change	94.0	91.6	94.0	90.9
Increase	6.0	8.4	5.4	9.1

Non-significant differences were found between all groups. PWV: pulse wave velocity, Δ: change, FU: follow-up, and BSL: baseline.

**Table 4 nutrients-17-00808-t004:** Multiple linear or logistic regression analysis between quartiles of sodium intake and changes in subclinical vascular biomarkers from baseline to 3 years of follow-up, using Q1 as the reference group.

	Model 1		Model 1a		Model 2		Model 3	
	B (95% CI)	*p*-Value	B (95% CI)	*p*-Value	B (95% CI)	*p*-Value	B (95% CI)	*p*-Value
ΔPWV (FU − BSL)								
Q2 vs. Q1	0.267 (−0.017, 0.551)	0.065	0.239 (−0.046, 0.523)	0.100	0.191 (−0.089, 0.471)	0.181	0.139 (−0.143, 0.421)	0.333
Q3 vs. Q1	0.066 (−0.220, 0.351)	0.652	0.090 (−0.195, 0.375)	0.535	0.048 (−0.234, 0.329)	0.739	−0.004 (−0.298, 0.290)	0.980
Q4 vs. Q1	0.110 (−0.193, 0.412)	0.477	0.142 (−0.157, 0.440)	0.351	0.044 (−0.254, 0.343)	0.770	−0.105 (−0.458, 0.249)	0.561
ΔPWV/Δt (FU − BSL)								
Q2 vs. Q1	0.079 (−0.010, 0.167)	0.080	0.071 (−0.018, 0.159)	0.119	0.054 (−0.033, 0.141)	0.221	0.038 (−0.049, 0.126)	0.390
Q3 vs. Q1	0.018 (−0.071, 0.107)	0.696	0.025 (−0.064, 0.114)	0.579	0.013 (−0.075, 0.100)	0.777	−0.004 (−0.095, 0.088)	0.938
Q4 vs. Q1	0.031 (−0.063, 0.125)	0.520	0.041 (−0.052, 0.134)	0.389	0.010 (−0.082, 0.102)	0.833	−0.036 (−0.146, 0.073)	0.516
	Model 1		Model 1a		Model 2		Model 3	
	Exp (B) (95% CI)	*p*-value	Exp (B) (95% CI)	*p*-value	Exp (B) (95% CI)	*p*-value	Exp (B) (95% CI)	*p*-value
Total plaques								
Q2 vs. Q1	0.580 (0.237, 1.416)	0.232	0.585 (0.241, 1.421)	0.236	0.595 (0.241, 1.466)	0.259	0.607 (0.209, 1.761)	0.358
Q3 vs. Q1	1.270 (0.592, 2.727)	0.539	1.375 (0.644, 2.935)	0.411	1.277 (0.593, 2.752)	0.532	1.412 (0.577, 3.454)	0.450
Q4 vs. Q1	0.829 (0.378, 1.818)	0.640	0.833 (0.381, 1.822)	0.648	0.837 (0.379, 1.848)	0.659	0.828 (0.353, 1.944)	0.665
Carotid plaques								
Q2 vs. Q1	0.801 (0.350, 1.835)	0.600	0.799 (0.352, 1.814)	0.592	0.769 (0.328, 1.804)	0.546	0.769 (0.278, 2.128)	0.613
Q3 vs. Q1	1.486 (0.708, 3.120)	0.295	1.505 (0.723, 3.131)	0.274	1.531 (0.727, 3.224)	0.262	1.585 (0.664, 3.781)	0.299
Q4 vs. Q1	0.624 (0.270, 1.440)	0.269	0.627 (0.272, 1.444)	0.273	0.645 (0.278, 1.496)	0.307	0.646 (0.264, 1.582)	0.339
Femoral plaques								
Q2 vs. Q1	0.579 (0.238, 1.408)	0.228	0.685 (0.286, 1.643)	0.397	0.538 (0.218, 1.329)	0.179	0.621 (0.215, 1.790)	0.377
Q3 vs. Q1	0.789 (0.346, 1.801)	0.574	0.950 (0.423, 2.134)	0.902	0.810 (0.351, 1.865)	0.620	0.925 (0.357, 2.394)	0.872
Q4 vs. Q1	0.533 (0.221, 1.284)	0.161	0.600 (0.251, 1.435)	0.251	0.507 (0.208, 1.234)	0.134	0.546 (0.214, 1.394)	0.206

**Model 1**: adjusted for age and sex. **Model 1a**: further adjusted for the interaction between inflammation and age. **Model 2**: adjusted for age, sex, existence of hypertension, existence of dyslipidemia, smoking, mean arterial pressure, and BMI. **Model 3**: adjusted for age, sex, existence of hypertension, existence of dyslipidemia, smoking, mean arterial pressure, BMI, inflammatory diseases, and energy intake. PWV: pulse wave velocity; FU: follow-up; BSL: baseline, Q: quartile; BMI: body mass index.

**Table 5 nutrients-17-00808-t005:** Percentages of misreporting of EI in total sample and quartiles of Na intake.

	Under-Report (%)	Normal (%)	Over-Report (%)
Total (n = 675)	49.6	45.9	4.4
Q1 (n = 169)	77.5	21.9	0.6
Q2 (n = 168)	58.3	41.1	0.6
Q3 (n = 170)	40.0	54.7	5.3
Q4 (n = 168)	22.6	66.1	11.3

EI: energy intake, and Na: sodium.

## Data Availability

Data can be provided after request.

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
