# Peer review of "Dietary Sodium Consumption and 3-Year Progression of Subclinical Arterial Damage in Adults with Cardiovascular Risk Factors"

_nutrients, 2025, doi:10.3390/nu17050808_

Round 1
Reviewer 1 Report
Comments and Suggestions for Authors
The manuscript addresses a relevant issue concerning the association between sodium consumption and the progression of subclinical arterial damage in a population with cardiovascular risk factors. The study is well structured and uses a longitudinal approach with a three-year follow-up, including a multivariate analysis to control for major confounding factors. However, there are some methodological and interpretive limitations that need to be addressed before the manuscript can be considered for publication.
One of the main problems concerns the estimation of sodium intake, which is based on two 24-hour dietary recalls. This method is known to be prone to memory errors and does not allow for accurate quantification of actual sodium intake, particularly with regard to sodium added to food during preparation or consumption. Urinary collection over 24 hours would be a more accurate method and would reduce the risk of misclassification. It would be useful for the authors to provide a more in-depth discussion of this limitation and the possible consequences on the results obtained.
Another critical point is the absence of a significant association between sodium consumption and the progression of subclinical arterial damage. Although this is an interesting result, it is not adequately discussed why the quartile with the highest sodium consumption did not show a negative effect, nor is the possibility of a non-linear relationship explored. Comparison with previous studies is useful, but the explanation of discrepancies with work that has found an association between sodium and arterial stiffness should be more thorough, taking into account differences in assessment methods, sodium levels consumed and characteristics of the population studied.
The follow-up duration of three years might be insufficient to detect significant changes in biomarkers of vascular damage, particularly plaque formation. This limitation is not adequately discussed and it would be useful to investigate how a longer observation period could influence the results. Furthermore, although the analysis included adjustment for several confounding factors, it is unclear whether significant interactive effects between sodium and other variables, such as age, gender or the presence of chronic inflammation, were explored.
A particularly critical aspect is the presence of similarty parts within the manuscript. Please check the MDPI file, as there are sections copied from the following article: https://www.mdpi.com/2072-6643/14/3/470/xml. This problem must be resolved before possible acceptance.
Author Response
Reply to Reviewer 1
Reviewer: “The manuscript addresses a relevant issue concerning the association between sodium consumption and the progression of subclinical arterial damage in a population with cardiovascular risk factors. The study is well structured and uses a longitudinal approach with a three-year follow-up, including a multivariate analysis to control for major confounding factors. However, there are some methodological and interpretive limitations that need to be addressed before the manuscript can be considered for publication.”
Reply to reviewer: We would like to thank the reviewer for the thoughtful suggestions that helped us improve the manuscript. Please find below a step-by-step reply to all comments. All the modified parts of the revised document are highlighted yellow.
“Comment”: One of the main problems concerns the estimation of sodium intake, which is based on two 24-hour dietary recalls. This method is known to be prone to memory errors and does not allow for accurate quantification of actual sodium intake, particularly with regard to sodium added to food during preparation or consumption. Urinary collection over 24 hours would be a more accurate method and would reduce the risk of misclassification. It would be useful for the authors to provide a more in-depth discussion of this limitation and the possible consequences on the results obtained.
Author’s response: We want to thank the reviewer for this comment. Indeed, 24-hour urinary collections are the gold-standard method for sodium intake assessment compared to other methods like 24-hour dietary recalls having important problems like recall bias and this was the reason that we also conducted a correction for misreporting. However, in order to be more accurate regarding this study’s limitations, we added some extra information in the limitations (line 321 – yellow highlighted sentence) and the manuscript changed as follows: “Furthermore, evaluation of Na intake was not conducted with the gold-standard method (24-hour urinary collection) but with two 24-hour recalls, which are based on participants memory and are prone to recall bias. Moreover, the use of 24-hour recalls do not allow the assessment of discretionary salt (either in cooking or while eating). In order to minimize any possible decline from the real intake, correction for misreporting has been conducted and results remained the same.”
“Comment”: Another critical point is the absence of a significant association between sodium consumption and the progression of subclinical arterial damage. Although this is an interesting result, it is not adequately discussed why the quartile with the highest sodium consumption did not show a negative effect, nor is the possibility of a non-linear relationship explored.
Author’s response: We thank the reviewer for this comment. In order to further explain the absence of associations in Q4, the following sentence was added (Discussion section, line 284): “As far as Q4 is concerned, the small duration of the follow-up that might be insufficient for vascular damage to appear, as well as the fact that participants were well-controlled (regulated lipid or BP levels and/or lower medication) patients, are the main reasons that might be responsible for the absence of relevant associations. Moreover, the absence of a significant association could also be attributed to the fact that participants in Q4 had better measurements as far as their lipid levels, medications and plaques are concerned, due to proper medical and lifestyle changes advice that were given after baseline measurements.” Finally, we tested both linear and non-linear regression associations between Na intake and vascular biomarkers, but the results remained not statistically significant.
“Comment”: Comparison with previous studies is useful, but the explanation of discrepancies with work that has found an association between sodium and arterial stiffness should be more thorough, taking into account differences in assessment methods, sodium levels consumed, and characteristics of the population studied.
Author’s response: Also, in respect to the reviewer’s comment, we added the following sentence regarding methodological differences between the current study and the studies available in the literature (Discussion section, line 268): “To sum up, there may be two available studies investigating similar associations in or-der to compare their results with the current study; however, there is great heterogeneity between all of them, since the current study (i) has the smallest study sample (n=675 vs n=1645 or n=1607), (ii) different methods of Na assessment were used in each study (24h recalls vs spot urine collections or FFQ & diet record), but not the gold standard method (24h urinary collection), (iii) differences in participants health status (adults with CVD risk factors vs CKD stage 3 patients or healthy adults), and (iv) the duration of follow-up in one of the studies (3 years vs 1 year).”
“Comment”: The follow-up duration of three years might be insufficient to detect significant changes in biomarkers of vascular damage, particularly plaque formation. This limitation is not adequately discussed and it would be useful to investigate how a longer observation period could influence the results. Furthermore, although the analysis included adjustment for several confounding factors, it is unclear whether significant interactive effects between sodium and other variables, such as age, gender or the presence of chronic inflammation, were explored.
Author’s response: We definitely agree with the reviewer’s comment. We have already added to the discussion the duration of the follow-up, which is short for plaques formation (in order to answer the above comment) and we also added a sentence in the limitations (line 328): “Last but not least, it should be mentioned that the follow-up duration may not sufficient for the detection of changes in plaques, since more time might be needed for plaque formation to occur.” Regarding the rest of the comment, it would be really interesting to find out about the influence of a longer period of time in the formation of plaques, and this is a reassurance that future research is needed. Moreover, we would like to inform the reviewer that interaction analyses were conducted between sodium intake and age, gender and existence of inflammation, as is already mentioned in the Methodology section (line 158). However, results were not statistically significant and are not presented.
“Comment”: A particularly critical aspect is the presence of similarty parts within the manuscript. Please check the MDPI file, as there are sections copied from the following article: https://www.mdpi.com/2072-6643/14/3/470/xml. This problem must be resolved before possible acceptance.
Author’s response: We want to thank the reviewer for the thorough inspection. Similar sections are due the methodology followed, which is written in the same way (since referring to the protocol) in many of our publications. However, in respect to the reviewers’ comment we have made several modifications in the Methodology section of this paper, to change the sentences as much as possible.

Reviewer 2 Report
Comments and Suggestions for Authors
This study examines the association of Na intake with markers of subclinical vascular disease and their change over time. No relationships were found.
It is of interest that the group with the highest Na intake had the lowest total cholesterol, the lowest LDL cholesterol levels, and the lowest prevalence of HTN/HTN meds and dyslipidemia at baseline, as well as plaques and carotid plaques. Conversely, participants in the lowest quartile of Na intake had the highest total cholesterol, the highest prevalence of diabetes, HTN/HTN drug use, PWV, and plaques. In other words, the distribution of baseline risk factors for subclinical vascular disease and the presence of subclinical vascular disease was not equally distributed among the quartiles of Na intake. Please comment.
Likewise, at follow up, the Q4 group had the lowest TC, LDL -C levels and the lowest prevalences of HTN/HTN med use, and dyslipidemia. Again, there is a mismatch between the groups in terms of risk factors for vascular disease. Please comment on this
Author Response
Reply to Reviewer 2
Reviewer: This study examines the association of Na intake with markers of subclinical vascular disease and their change over time. No relationships were found.
Reply to reviewer: We would like to thank the reviewer for their comments about our manuscript. Please find below a point-by-point response.
“Comment”: It is of interest that the group with the highest Na intake had the lowest total cholesterol, the lowest LDL cholesterol levels, and the lowest prevalence of HTN/HTN meds and dyslipidemia at baseline, as well as plaques and carotid plaques. Conversely, participants in the lowest quartile of Na intake had the highest total cholesterol, the highest prevalence of diabetes, HTN/HTN drug use, PWV, and plaques. In other words, the distribution of baseline risk factors for subclinical vascular disease and the presence of subclinical vascular disease was not equally distributed among the quartiles of Na intake. Please comment.
Author’s response: We thank the reviewer for this comment. Indeed, participants in Q4 (highest Na intake) had better measurements as far as their lipid levels, medications and plaques are concerned. Despite the fact that we adjusted for several confounding factors (i.e. age, sex, existence of hypertension, existence of dyslipidemia, smoking, mean arterial pressure, BMI, inflammatory diseases and energy intake), the effect of these factors might not have totally disappeared. Moreover, we should also mention the reverse causality effect that might be responsible for this result, since all of the participants were adults with CVD risk factors and might have taken medical advice to modify their lifestyle, so that they would help their health status. This limitation has now been added to the limitations (line 317 – yellow highlighted): “However, it also has some limitations, due to its cross-sectional design, no causative relationships can be established, and the reverse causality phenomenon might be the reason for the very well controlled participants (i.e. lipid profile, medication) in the quartile with the highest Na intake (Q4).”
“Comment”: Likewise, at follow up, the Q4 group had the lowest TC, LDL -C levels and the lowest prevalences of HTN/HTN med use, and dyslipidemia. Again, there is a mismatch between the groups in terms of risk factors for vascular disease. Please comment on this
Author’s response: We appreciate the reviewer for this comment. As in the previous comment, the main reason for this result is the reverse causality effect. Participants at baseline conducted several measurements, based on which they were given proper medication and advice for changes in their lifestyle. This is the main reason that after 3 years (at follow-up), they had already made some lifestyle modifications and taken their medication (if needed), that resulted in a more controlled health status than participants in other quartiles. For this reason, as it has already been mentioned in the previous comment, the following yellow highlighted sentence has been added to the study’s limitations (line 317): “However, it also has some limitations, due to its cross-sectional design, no causative relationships can be established, and the reverse causality phenomenon might be the reason for the very well controlled participants (i.e. lipid profile, medication) in the quartile with the highest Na intake (Q4).”

Round 2
Reviewer 1 Report
Comments and Suggestions for Authors
The manuscript has made progress, but there are still methodological and interpretive issues that need further revision before publication.
The estimation of sodium intake through two 24-hour dietary recalls remains an important limitation. Although the authors discussed this critical issue in the limitations section and made a correction for misreporting, the method used does not allow for accurate quantification of discretionary sodium. It would be useful to further investigate the possible impact of this limitation on the results.
The absence of a significant association between sodium consumption and progression of subclinical arterial damage is interesting, but the explanation for the lack of effect in the highest quartile of sodium intake remains unclear. The discussion improved with the addition of hypotheses regarding the duration of follow-up and control of risk factors in the participants, but a more in-depth analysis of the possible non-linear relationship (as well as testing for interactions with other factors such as age and inflammation) would be helpful to strengthen the conclusions.
The comparison with the literature was expanded to include a more detailed discussion of methodological differences between the studies. However, the explanation of discrepancies with studies that found an association between sodium and arterial damage could be further developed, also taking into account the potential effect of a different overall dietary composition.
Author Response
Reply to Reviewer 1
Reviewer: “The manuscript has made progress, but there are still methodological and interpretive issues that need further revision before publication.”
Reply to reviewer: We want to thank the reviewer for all their suggestions, that help the understanding and improvement of the manuscript. Please find below a step-by-step reply to all comments. All the modified parts of the revised document are highlighted yellow.
“Comment”: The estimation of sodium intake through two 24-hour dietary recalls remains an important limitation. Although the authors discussed this critical issue in the limitations section and made a correction for misreporting, the method used does not allow for accurate quantification of discretionary sodium. It would be useful to further investigate the possible impact of this limitation on the results.
Author’s response: We agree with the reviewer’s comment regarding the difficulties in quantification of discretionary sodium. Lack of accurate methods of dietary intake assessment has been widely discussed in the past. As far as 24-hour dietary recall is concerned, some believe that the more dietary recalls we use the better, as depicted in some studies (1) supporting the use of 8 repeated 24hr-recalls to accurately capture habitual dietary intake. However, we should keep in mind that in large population studies even one 24hr-recall is considered appropriate in estimating mean nutrient intake (2) and it has also been shown that a second 24-hr recall even in a different season does not confer additional information (3). One should also keep in mind that dietary assessment is a laborious work mainly for the study participants and repeated 24hour-recalls may lead to severe drop-outs even in highly motivated volunteers. More importantly repeated 24hr-recalls may influence eating habits of the participants leading to serious bias in the dietary assessment.
In 2002 the European Project: European Food Consumption Survey Method (EFCOSUM) (4), aimed to develop a common basic method for dietary assessment that should be used in the forthcoming nutrition studies, allowing for reliable comparisons. Conclusions of this project were: " Since the 24-h recall method is applicable in large European populations of different ethnicity, has a relatively low respondent and interviewer burden, is open-ended and is cost-effective, this method can be considered as the best method for EFCOSUM to get population mean intakes and distributions for subjects aged 10 y and over in different European countries. Usual intake should be estimated by statistical modelling techniques, using two non-consecutive 24 h-recalls and a food list to assess the proportion non-users for infrequently consumed foods".
As it comes to sodium intake, assessment becomes even harder. According to published data discretionary salt, which is not assessed through any dietary intake assessment method is estimated to be around 15% of the total daily sodium intake (5,6). In large epidemiological studies (e.g. NHANES) sodium intake is being assessed through 24-h dietary recalls (7).
In line with the above-mentioned data, we decided to use 24-hr recalls of two non-consecutive days as one of the best methods for dietary assessment, avoiding dropouts and assessment bias. However, as already depicted in the beginning of the present answer, we recognize the possible bias of misreporting, which is present almost equally in all available dietary assessment methods. In order to minimize the impact of this limitation as much as possible we added 15% of the amount of sodium consumed and corrected for misreporting. To our knowledge, until today, the estimation of discretionary salt in studies remains a challenge for the investigators, and there is no generally accepted protocol to be applied in dietary surveys. We are willing to further investigate the impact of this limitation on the results if the reviewer has any suggestions for us that we might have neglected.
1. KA Jackson, NM Byrne, AM Magarey and AP Hills. Minimizing random error in dietary intakes assessed by 24-h recall, in overweight and obese adults. Eur J Clin Nutr 2008 62, 537–543.
2. McPherson, R.S., Hoelscher, D.M., Alexander, M., Scanlon, K.S. & Serdula, M.K. Dietary assessment methods among school-aged children: validity and reliability. Prev Med 2000 31, S11–S33.
3. M. Yannakoulia, A. C. Drichoutis, M. D. Kontogianni & F. Magkanari. Season-related variation in dietary recalls used in a paediatric population. J Hum Nutr Diet, 2010; 23: 489–493.
4. G. Biro, K.F.A.M. Hulshof, L. Ovesen and J.A. Amorim Cruz for the EFCOSUM Group. Selection of methodology to assess food intake. Eur J Clin Nutr 2002 56, S25–S32.
5. Elliott P & Brown I (2007) Sodium Intakes around the World. Geneva: World Health Organization; available at http:// www.who.int/iris/handle/10665/43738.
6. James WP, Ralph A & Sanchez-Castillo CP (1987) The dominance of salt in manufactured food in the sodium intake of affluent societies. Lancet 1, 426–429.
7. Hu, J.R., et al., Dietary Sodium Intake and Sodium Density in the United States: Estimates From NHANES 2005-2006 and 2015-2016. Am J Hypertens, 2020. 33(9): p. 825-830.
“Comment”: The absence of a significant association between sodium consumption and progression of subclinical arterial damage is interesting, but the explanation for the lack of effect in the highest quartile of sodium intake remains unclear. The discussion improved with the addition of hypotheses regarding the duration of follow-up and control of risk factors in the participants, but a more in-depth analysis of the possible non-linear relationship (as well as testing for interactions with other factors such as age and inflammation) would be helpful to strengthen the conclusions.
Author’s response: We thank the reviewer for the suggestions. Regarding the absence of associations in Q4 the paragraph that further analyzes the reasons was changed to the following and can be found in Discussion section (line 287): “The fact that the highest quartile did not show any statistically significant change in vascular biomarkers, could be also attributed to the combination of the following reasons: (i) mean sodium intake in this specific quartile was 3487.92±1025.92 mg Na/day, which is almost twice the recommended intake, but is not an exaggerated amount (mean worldwide intake is about 4300 mg/day), (ii) participants in Q4 had better measurements as far as their lipid levels, medications and plaques are concerned, due to proper medical and lifestyle changes advice that were given after baseline measurements, (iii) participants in Q4 were the youngest (mean age in follow-up:49.43±13.03 years)of the total sample, (iv) had the lowest mean PWV of the total sample (7.96±2.23 m/sec) and (v) the short duration of the follow-up that might be insufficient for vascular damage to appear. Consequently, the possible reduction in sodium intake that participants with might have been advised to follow, their age that was between 45 - 50 years until the follow-up, where the normal increase in PWV is slow (and in plaques is even slower) [1], and the fact that the follow-up was not conducted years later, are a very reasonable explanation of the absence of associations.”
As far as interaction analyses with factors such as age, gender, inflammation, potassium are concerned, they have already been conducted and results were not statistically significant, and this is the reason for not presenting them. However, a reference to every interaction analysis that was conducted can be found in Methodology section (line 158). Furthermore, adjustment for total energy intake was also conducted in linear and logistic regression analyses (Table 4).
“Comment”: The comparison with the literature was expanded to include a more detailed discussion of methodological differences between the studies. However, the explanation of discrepancies with studies that found an association between sodium and arterial damage could be further developed, also taking into account the potential effect of a different overall dietary composition.
Author’s response: We totally understand the reviewer’s point and we also agree, but the existing studies do not analyze their participants’ dietary composition, in order to compare the different effects that might occur. However, since it could really be a very reasonable explanation, we added a relevant sentence in the Discussion section (line 268): “Last but not least, an important point that might be the reason for the differences between this study and the available studies, is the overall dietary composition of each study’s participants; however, these information are not available in order to verify this possibility.”

Reviewer 2 Report
Comments and Suggestions for Authors
the paper is fine
Please change "small duration" to "short duration"
Author Response
Reply to Reviewer 2
Reviewer: “the paper is fine”.
Reply to reviewer: We would like to thank the reviewer for the helpful comments that improved our manuscript.
“Comment”: Please change "small duration" to "short duration"
Author’s response: Thank you for this. The change can be found on line 296.
